# Microdroplets initiate organic-inorganic interactions and mass transfer in thermal hydrous geosystems

Guanghui Yuan [1,2] ✉, Zihao Jin[1], Yingchang Cao [1] ✉, Hans-Martin Schulz [3], Jon Gluyas [4], Keyu Liu [1], Xingliang He [5] & Yanzhong Wang[1]

Organic-inorganic interactions regulate the dynamics of hydrocarbons, water, minerals, $CO_2$, and $H_2$ in thermal rocks, yet their initiation remains debated. To address this, we conducted isotope-tagged and in-situ visual thermal experiments. Isotope-tagged studies revealed extensive H/O transfers in hydrous $n$-$C_{20}H_{42}$-$H_2O$-feldspar systems. Visual experiments observed water microdroplets forming at 150–165 °C in oil phases near the water-oil interface without surfactants, persisting until complete miscibility above 350 °C. Electron paramagnetic resonance (EPR) detected hydroxyl free radicals concurrent with microdroplet formation. Here we propose a two-fold mechanism: water-derived and $n$-$C_{20}H_{42}$-derived free radicals drive interactions with organic species, while water-derived and mineral-derived ions trigger mineral interactions. These processes, facilitated by microdroplets and bulk water, blur boundaries between organic and inorganic species, enabling extensive interactions and mass transfer. Our findings redefine microscopic interplays between organic and inorganic components, offering insights into diagenetic and hydrous-metamorphic processes, and mass transfer cycles in deep basins and subduction zones.

Organic–inorganic interactions are ubiquitous in deep sedimentary basins, subduction zones and hydrothermal vents, governing the behaviors and fate of hydrocarbons, water, minerals, $CO_2$, and $H_2$[1–6]. Diverse catalytic models, such as Fischer–Tropsch-type reactions in serpentinization[1,6–9], the catalytic influence of clay minerals on kerogen maturation[4,10], and the catalysis of iron-bearing minerals[4,11–13] or sulfate minerals[14,15] on hydrocarbon evolution, have been proposed to elucidate these interactions in natural geochemical systems. These models are particularly effective in environments where specific catalyst minerals are present.

The transfer of water-derived H/O to kerogen, newly formed hydrocarbons[16–21] and oxygenated compounds[22,23], as well as transfer of alkane-derived hydrogen to newly formed water molecules[21], have

been observed in thermal experiments. Notably, this H/O transfer, involving the dissociation and regeneration of water molecules, has also been identified in some thermal experiments without catalysts[18,21]. This suggests the existence of a non-catalytic mechanism that may initiate interactions between organic species and hot water.

Different from bulk water, water microdroplets at room temperature exhibit unique behavior[24,25]. They produce hydroxyl radicals and hydrogen peroxide[25–31], facilitating the acceleration of organic reactions[32–34], all without requiring additional catalysts. Chen et al.[27] and Song et al.[30] also reported the initiation of accelerated alkane degradation by contact of different alkanes with man-made water microdroplets at room temperature. However, non-catalytic interactions in thermal hydrous systems are still a matter of debate in the

[1]State Key Laboratory of Deep Oil and Gas, China University of Petroleum (East China), Qingdao, P.R. China. [2]Institute of Energy, School of Earth and Space Sciences, Peking University, Beijing, P.R. China. [3]GFZ German Research Centre for Geosciences, Telegrafenberg, Potsdam, Germany. [4]Department of Earth Sciences, Durham University, Durham, UK. [5]Qingdao Institute of Marine Geology, China Geological Survey, Qingdao, China.
✉ e-mail: yuan.guanghui@upc.edu.cn; caoych@upc.edu.cn

geochemical community[5,35]. Two schools of thought exist regarding this matter. One group suggests that as water ionizes to generate more $H^+$ and $OH^-$ at elevated temperature[36], interactions between hot water and organic species proceed via ionic mechanisms[36–39]. However, the hypothetical preference for ionic mechanisms contradicts the dominance of straight-chain alkanes observed in hydrous pyrolysis experiments[5,17,19] and natural mature petroleum and light hydrocarbons[40–42]. Another group proposes the free radical mechanism[17,19], favoring the generation of straight-chain alkanes through water-derived and hydrocarbon-derived free radicals. Lewan[17] referred Hostettler[43] and proposed the formation of hydrogen radicals in hot water following the reaction of $H_2O_{(aq)} + e^-_{(aq)} \rightleftharpoons OH^-_{(aq)} + H^*_{(aq)}$. However, bulk hot water at 300–400 °C is not likely to generate free radicals without radiolysis[38,43]. Furthermore, the potential transfer of hydrocarbon-derived H to OH-containing minerals and aluminosilicate mineral-derived O to other species (e.g., $CO_2$) remains poorly understood, leaving gaps in our understanding of interactions among various organic and inorganic species in thermal geochemical systems.

In the realm of deeply buried, low-permeability rocks housing diverse geofluids, the formation of microdroplets becomes feasible, especially in high-temperature, high-pressure (HTHP) environments[44]. These microenvironments may host intricate interactions among diverse organic compounds, water, minerals, and gases, with significant geological implications often overlooked[27,30,32]. Motivated by recent advances in interfacial chemistry[24–27,30,32], we aim to investigate the formation, evolution, and characteristics of water microdroplets near alkane–water interfaces at elevated temperatures, and to explore the basic physicochemical processes underpinning the organic–inorganic interactions in thermal geochemical systems from a microscopic perspective. Our approach involves isotope-tagged experiments in HTHP Hastelloy reactors (Supplementary Fig. 1) and in situ visual experiments in transparent fused silica capillary tubes (FSCTs) (Supplementary Fig. 2). Employing three compounds—$n$-$C_{20}H_{42}$($n$-$C_{20}D_{42}$), $H_2O$ ($D_2O$, $D_2^{18}O$), and K-feldspar—we explore chemical reactions and H/O transfer across different species. $n$-Eicosane, a key petroleum constituent, water, and feldspar are chosen for their relevance in hydrogeochemical reactions and natural prevalence. Our comprehensive approach involves ten sets of experiments in Hastelloy reactors, allowing for diverse combinations of compounds (Supplementary Table 1 and Supplementary Fig. 1). In the in situ visual experiments (Supplementary Fig. 2), we utilize $H_2O$ and three types of oils ($n$-$C_{20}H_{42}$, the liquid hydrocarbon produced from the pyrolysis of $n$-$C_{20}H_{42}$ with $H_2O$, and a deep crude oil from Bohai Bay Basin, East China (Supplementary Fig. 3) to mirror processes in Hastelloy reactors and demonstrate universality of microdroplet generation in both HTHP reactors and deep hydrocarbon reservoirs. The temperature ranges for visual experiments span from 25 °C to 360–410 °C, corresponding to the point of complete miscibility between water and different oils (Supplementary Table 2). In addition, electron paramagnetic resonance (EPR) tests are performed to detect the hydroxyl free radicals produced from water microdroplets formed in FSCTs.

## Results and discussion
### Water microdroplets and hydroxyl radicals they generate
In our in situ visual thermal experiments, we observed a distinct interface between the bulk water and $n$-$C_{20}H_{42}$ phases at 25 °C in the first FSCT system. Below 150 °C, no discernible formation of water or oil microdroplet was evident near the interface (Fig. 1a1 and Supplementary Movie 1). However, at 150 °C, after a 10-min period, small water microdroplets became identifiable in the $n$-$C_{20}H_{42}$ phase adjacent to the interface (Fig. 1a2 and Supplementary Movie 2). The experiment progressed as numerous water microdroplets, each smaller than 5 μm, formed and gradually merged, coalescing into larger units ranging from 20 to 30 μm (Supplementary Movies 3 and 4).

Subsequently, these large water droplets collapsed, reverting back into small microdroplets. These coalescence and bursting events occurred cyclically, with each cycle lasting less than 1–2 min. As the temperature increased, the formation and evolution of water microdroplets became more extensive (Fig. 1a3, a4 and Supplementary Movies 4 and 5), with the duration of each coalescence-bursting cycle reduced to less than 10 s. In the $n$-$C_{20}H_{42}$-water system, only a few $n$-$C_{20}H_{42}$ microdroplets formed until the temperature reached 360 °C (Fig. 1a5). As the temperature further increased, $n$-$C_{20}H_{42}$ microdroplets formed extensively (Fig. 1a6), coalescing into large droplets up to 40 μm, which then burst back into smaller microdroplets. Even at 400 °C, the interface between the bulk water and $n$-$C_{20}H_{42}$ phases remained but exhibited substantial exchange of $n$-$C_{20}H_{42}$ and water across it (Fig. 1a6). Finally, at 402 °C, the interface disappeared with complete miscibility of the $n$-$C_{20}H_{42}$ and water.

In the second FSCT system containing water and liquid hydrocarbon obtained from pyrolyzed $n$-$C_{20}H_{42}$, a distinct interface also existed between the bulk water and hydrocarbon phases. No water microdroplet was observed near the interface until the temperature reached 150 °C (Fig. 1b1 and Supplementary Movie 6). At 150 °C, after a minute of heating, small water microdroplets started to form in the hydrocarbon phase near the interface (Fig. 1b2 and Supplementary Movie 7). The subsequent processes of water microdroplet formation and evolution in this system were similar to those in the $n$-$C_{20}H_{42}$-water system (Fig. 1b3, b4 and Supplementary Movie 8). However, hydrocarbon microdroplets began forming at 330 °C (Fig. 1b5 and Supplementary Movie 9), which was 30 °C lower than the 360 °C observed in the $n$-$C_{20}H_{42}$-water system. The interface between the bulk water and hydrocarbon phases still existed at 348 °C (Fig. 1b6), but with extensive exchange of both hydrocarbon and water across it. Finally, the interface vanished as the hydrocarbon and water became complete miscible at 352 °C, a much lower temperature than the 402 °C observed in the $n$-$C_{20}H_{42}$ and water system.

In the third FSCT system with water and crude oil, a clear interface separated the bulk water and oil phases. No water microdroplet was observed until the temperature reached 165 °C (Fig. 1c1, c2 and Supplementary Movies 10, 11), which was 15 °C higher than observed in the other two systems. From this point on, the formation and evolution of water microdroplets in this system resembled that observed in the other two systems (Fig. 1c3 and Supplementary Movie 12). However, a substantial amount of small oil microdroplets began forming in the water phase near the interface at 200 °C (Fig. 1c4 and Supplementary Movie 13), a much lower temperature than required in the other two systems. These oil microdroplets also underwent coalescing-bursting cycles, similar to the behavior of water microdroplets (Fig. 1c5 and Supplementary Movies 14 and 15). The interface between the bulk water and oil phase persisted at 380–390 °C (Fig. 1c6), but there was extensive exchange of both hydrocarbon and water microdroplets across it. Finally, at 393 °C, the interface disappeared as the oil and water became completely miscible.

While microdroplets typically form in oil–water systems with the presence of surfactants[45] or under the influence of ultrasonication[27], our in situ visual experiments demonstrate that water and oil microdroplets can form near interfaces in alkane/oil–water systems without surfactants, but at a temperature higher than 150 °C, and persisting clearly until complete miscibility of oil and water phases at a temperature exceeding 350 °C. Within 300–1000 μm thick zones on both sides of the interface, water or alkane (oil) microdroplets were observed to migrate, with their abundance decreasing significantly away from the interface. Recent studies at room temperature have shown that water microdroplets smaller than 15 μm can generate hydroxyl radicals (OH*) at the water–alkane (air) interfaces[25,29,32]. Specifically, Chen et al. observed the formation of H* and OH* in a hexadecane-water emulsion system with numerous water microdroplets created via ultrasonification at room

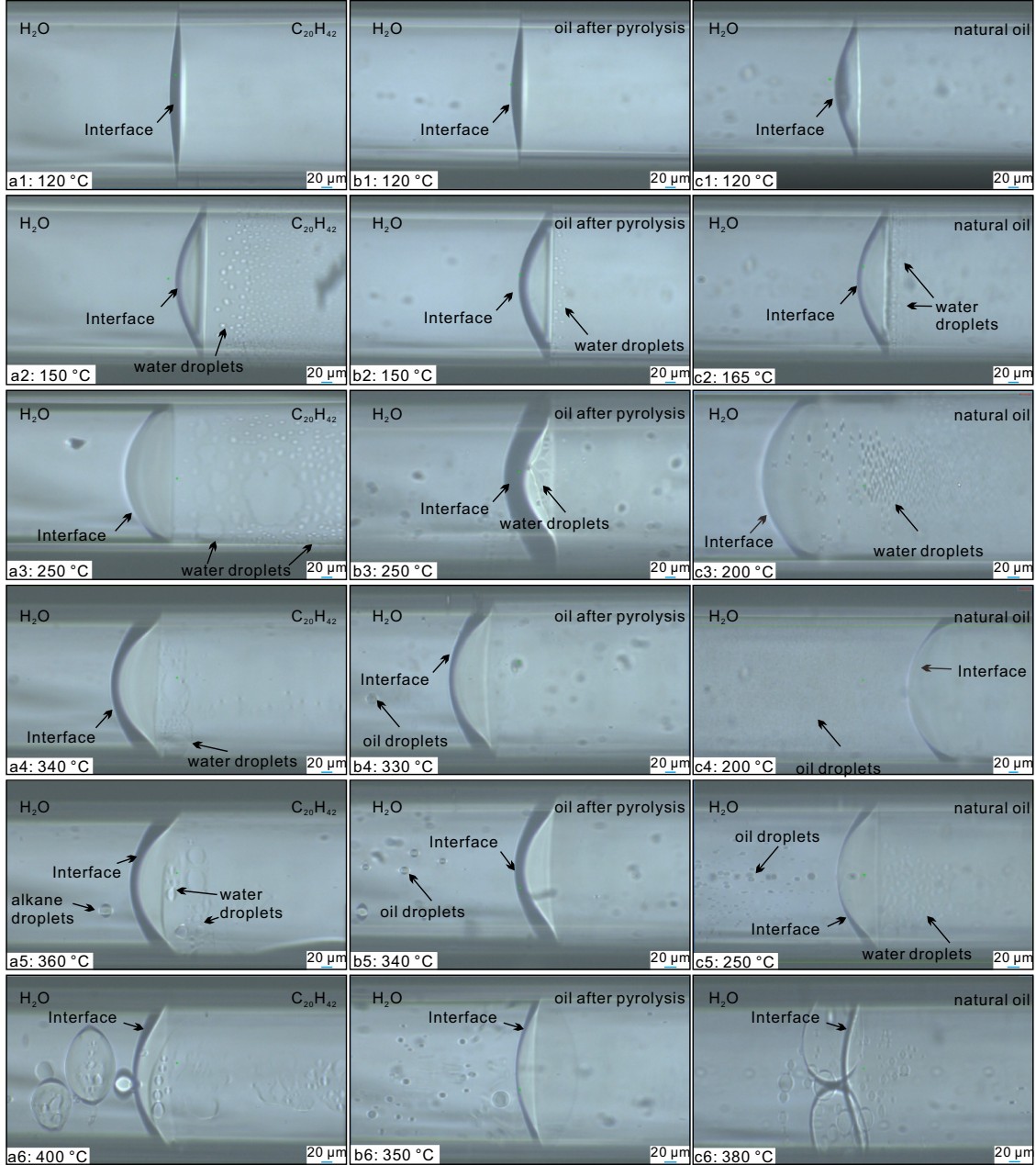

**Fig. 1 | Formation and evolution of water/oil microdroplets near the interfaces from low to high temperatures in three different FSCT systems. a1–a6** microdroplets evolution in a system with water and $n$-$C_{20}H_{42}$. **b1–b6** microdroplets evolution in a system with water and liquid hydrocarbon from pyrolyzed $n$-$C_{20}H_{42}$. **c1–c6** microdroplets evolution in system with water and crude oil. Microdroplets do not form at low temperatures. However, when the temperatures exceed

150–165 °C, high temperature and high pressure facilitate the creation of numerous water microdroplets at the oil–water interface. These microdroplets range in size from 5 μm to 30 μm and undergo continuous dynamic evolution. Small microdroplets converge into large microdroplets, which then burst to generate small microdroplets. These microdroplet persist until complete miscibility is achieved between water and oil at temperatures exceeding 350 °C.

temperature (see Fig. 1c in ref. 27). Following our in situ visual experiments, we conducted EPR spectra tests on different oil–water systems below and above the critical temperature. Hydroxyl radicals were clearly detected in all FSCTs containing water and three different alkane/oils at 200 °C when water microdroplets were extensively formed (Fig. 2a–c), with more than ten oil–water interfaces present in the FSCTs. However, at 140 °C, no signal of hydroxyl radicals was observed (Fig. 2a). Thus, our tests demonstrate that water-derived hydroxyl radicals can also be formed at elevated temperatures following the formation of water microdroplets near oil–water interfaces, even without radiation.

## Organic–inorganic interactions in anhydrous and hydrous systems

In our Hastelloy reactor experiments, significant differences were observed in minerals, gases and liquid hydrocarbons between anhydrous to hydrous systems. In the anhydrous system with only $n$-$C_{20}H_{42}$ (Experiment I), the primary gases formed via $n$-$C_{20}H_{42}$ degradation were $H_2$ (0.550 ml/g), ethane ($C_2H_6$, 0.187 ml/g), and propane ($C_3H_6$, 0.114 ml/g), followed by methane ($CH_4$, 0.078 ml/g), $n$-butane ($n$-$C_4H_{10}$, 0.034 ml/g), and traces of ethene ($C_2H_4$), propene ($C_3H_6$) and pentane ($C_5H_{12}$) (Table 1 and Fig. 3a). The gas chromatograms (GC) of post-reaction liquid hydrocarbons show that $n$-$C_{20}H_{42}$ evolved to form

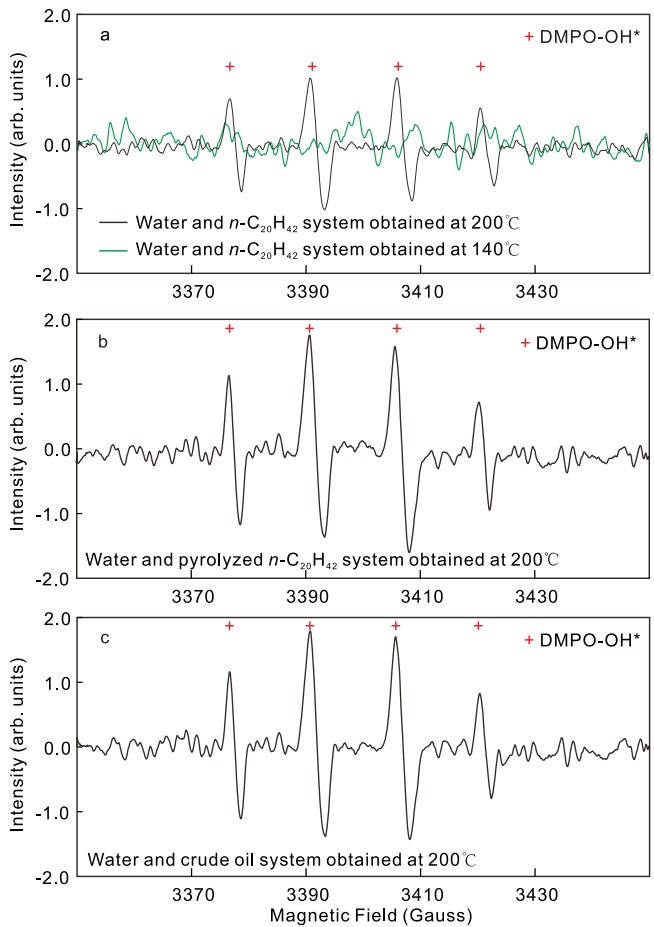

**Fig. 2 | Electron paramagnetic resonance (EPR) spectra of hydroxyl free radicals (OH\*) obtained in three different water–oil systems, with 5,5-dimethyl-1-pyrroline N-oxide (DMPO) as the probe. a** EPR spectra obtained at 200 °C and 140 °C in system with water and $n$-$C_{20}H_{42}$. **b** EPR spectra obtained at 200 °C in system with water and liquid hydrocarbon from pyrolyzed $n$-$C_{20}H_{42}$. **c** EPR spectra obtained at 200 °C in system with water and crude oil. Detailed data have been deposited in Figshare.

both low-molecular-weight (LMW) organics (mainly $nC_6$-$nC_{19}$) and some high-molecular-weight (HMW) organics ($nC_{17}$-$nC_{30}$ and some polycyclic aromatic hydrocarbons) (Fig. 3b1). In the anhydrous $n$-$C_{20}H_{42}$-feldspar systems (Experiment-II), feldspar grains remained unaltered (Fig. 4a, b). The introduction of feldspar slightly increased the yields of ethane (0.273 ml/g), propane (0.150 ml/g) and $nC_6$-$nC_{13}$, while reducing the $H_2$ yield to 0.491 ml/g (Table 1 and Fig. 3a, b2). Yields of other gases or liquid hydrocarbons showed minimal difference compared to Experiment I. No $CO_2$ was detected in these anhydrous systems. These observations suggest weak interactions between $n$-$C_{20}H_{42}$ and feldspar in anhydrous systems.

In contrast, extensive interactions between organic and inorganic components occur in hydrous systems. Unlike the anhydrous system, notable feldspar dissolution occurred in the hydrous systems, accompanied by the precipitation of secondary minerals (Fig. 4c–l). In the hydrous systems containing 20 mg feldspars (III, IV), feldspar grains underwent extensive dissolution (Fig. 4c–f), with some secondary minerals, including boehmite and kaolinite appearing on the feldspar surfaces. In systems with 2 g feldspars (V, VII, VIII), the feldspar grains were also leached, with simultaneously widespread precipitation of secondary minerals such as illite and muscovite and some boehmite on most feldspar surfaces (Fig. 4h–l). In addition, the presence of water ($H_2O$, $D_2O$) significantly promoted the degradation of $n$-$C_{20}H_{42}$ to generate more methane (0.531–0.557 ml/g), ethane

(0.847–0.963 ml/g), propane (0.551–0.631 ml/g), $n$-butane (0.072–0.141 ml/g) in Experiment-III ($n$-$C_{20}H_{42}$ + $H_2O$ + 20 mg feldspar) and Experiment-IV ($C_{20}H_{42}$ + $D_2^{16}O$ + 20 mg feldspar), while the $H_2$ yield decreased to 0.338–0.427 ml/g. Besides gases, much more LMW liquid alkanes were produced in the hydrous systems, while the amount of HMW hydrocarbons increased little (Fig. 3b3–b4). Moreover, the introduction of $^{18}O$-labeled $D_2^{18}O$ (Experiment-V $C_{20}H_{42}$ + $D_2^{18}O$ + 2 g feldspar, Experiment-VI $C_{20}H_{42}$ + $D_2^{18}O$) resulted in substantially higher gas generation, particularly increased yields of methane, ethane, propane and $H_2$ (Table 1 and Fig. 3a). Another significant difference from anhydrous systems was the production of $CO_2$ (Fig. 3a and Table 1) and organic acids (see Fig. 7D in ref. 5) in the hydrous systems. In addition, the presence of feldspar resulted in higher yields of hydrocarbon gases, $H_2$ and $CO_2$ in Experiment-III (compared with Experiment-X) and Experiment V (compared with Experiment-VI), regardless of the water type (Table 1 and Fig. 3a).

## Mass transfer of H/O in $n$-$C_{20}H(D)_{42}$-water-feldspar system

In our anhydrous $n$-$C_{20}H_{42}$-(feldspar) systems (Experiment I, II), where an additional hydrogen source was absent, $n$-$C_{20}H_{42}$ exclusively provided hydrogens for the newly formed gaseous and liquid hydrocarbons. However, in our hydrous systems, the δD compositions of gaseous and liquid hydrocarbons exhibit significant differences between the $n$-$C_{20}H_{42}$ + $H_2^{16}O$ + 20 mg feldspar system (III) and the $C_{20}H_{42}$ + $D_2^{16}O$ + 20 mg feldspar system with D-labeled water (IV) (Fig. 5a). Without $D_2^{16}O$, the δD (AT D/H) values of $CH_4$ and $C_2H_6$ are −287‰ (0.0111) and −313‰ (0.0107), respectively, and the values of liquid hydrocarbons range from −162‰ to −33‰ (0.0130–0.0151) (Fig. 5a and Supplementary Table 3). With D-labeled water, the δD (AT D/H) values of $CH_4$ and $C_2H_6$ are 938,607‰ (12.7661) and 185,123‰ (2.8172), respectively, and the values of liquid hydrocarbons range from 22,027‰ (0.3574) to 14,382‰ (0.2390) (Fig. 5a and Supplementary Table 3). Particularly, the δD (AT D/H) values of $n$-$C_{20}H_{42}$ after experiment also increased significantly to 5182‰ (0.0962) where $D_2O$ was present (Fig. 5a). In addition to the isotopic compositions, the high-field nuclear magnetic resonance (HF-NMR) spectra also show distinct differences in the systems without and with D-labeled water (Fig. 6 and Supplementary Fig. 4). We obtained notable deuterium signals in the HF-NMR spectra of the newly formed liquid alkanes (signals at 0.5–1.7 ppm), ketones (1.7–2.5 ppm), and hydrocarbons containing benzene rings (6.7–7.0 ppm) in the system with D-labeled water (Experiment V, VI) (Fig. 6a). In experiments III ($C_{20}H_{42}$ + $H_2^{16}O$ + 20 mg feldspar) and X ($C_{20}H_{42}$ + $H_2^{16}O$), however, no deuterium signal was detected in the liquid hydrocarbons (Fig. 6a). The considerably higher δD values of various alkanes and the distinct NMR deuterium signals demonstrate the incorporation of water-derived hydrogen into the newly formed gases, liquid hydrocarbons and other organic components. The $n$-$C_{20}H_{42}$ and $n$-alkane products are devoid of hydrophilic NSO groups, making the hydrogen in them less likely to undergo low-temperature isotopic exchange with water-derived H[20]. Research indicates that hydrogen exchange between $CH_4$ and water in $CH_4$-water systems (323 °C) is considerably slower than in $C_2$–$C_5$ alkane–water systems[12]. The higher δD value of $CH_4$ compared to $C_2H_6$ obtained in the $n$-$C_{20}H_{42}$-$D_2O$ system (IV) (Fig. 5a and Supplementary Table 3) suggests that most of the C–D bonds in the newly formed hydrocarbons were directly established during the degradation processes of $n$-$C_{20}H_{42}$, rather than through subsequent hydrogen exchange between the formed alkanes and water. Hence, both $n$-$C_{20}H_{42}$ and water contribute hydrogen to the newly formed gaseous and liquid hydrocarbons in the $n$-$C_{20}H_{42}$-water-(feldspar) systems.

In our anhydrous $n$-$C_{20}H_{42}$-feldspar system (II), the absence of $CO_2$ generation at 340 °C indicates that oxygen in feldspar does not oxidize carbon in $n$-$C_{20}H_{42}$. However, in the hydrous systems, $n$-$C_{20}H_{42}$ degradation is associated with the production of $CO_2$ (Fig. 3a and Table 1) and oxygen-containing organics[4,5]. In experiments III and IV

**Table 1 | Gas yields of $C_1$–$C_5$, $H_2$, and $CO_2$ in different thermal experiments after 14-d heating**

| Gases | | Experiment No. | | | | | | | | | |
|---|---|---|---|---|---|---|---|---|---|---|---|
| | | I | II | III | IV | V | VI | VII | VIII | IX | X |
| Yields (ml/g) | $C_1$ | 0.078 | 0.088 | 0.531 | 0.557 | 5.742 | 1.46 | 0.323 | 0.117 | / | 0.28 |
| | $C_2$ | 0.187 | 0.273 | 0.963 | 0.847 | 6.479 | 1.804 | 0.497 | 0.362 | / | 0.683 |
| | $C_{2ene}$ | 0.002 | 0.003 | 0.004 | 0.003 | 0.031 | 0.013 | 0.006 | 0.004 | / | 0.003 |
| | $C_3$ | 0.114 | 0.15 | 0.551 | 0.631 | 2.698 | 0.726 | 0.268 | 0.199 | / | 0.367 |
| | $C_{3ene}$ | 0.017 | 0.027 | 0.031 | 0.025 | 0.153 | 0.061 | 0.029 | 0.036 | / | 0.031 |
| | $i$-$C_4$ | 0 | 0 | 0.001 | 0.002 | 0.02 | 0.001 | 0 | 0 | / | 0.003 |
| | $n$-$C_4$ | 0.034 | 0.038 | 0.141 | 0.072 | 0.563 | 0.176 | 0.072 | 0.051 | / | 0.107 |
| | $i$-$C_5$ | 0.002 | 0.002 | 0.003 | 0 | 0.008 | 0.002 | 0.003 | 0.003 | / | 0.003 |
| | $n$-$C_5$ | 0.002 | 0.003 | 0.001 | 0.002 | 0.006 | 0.003 | 0.004 | 0.004 | / | 0.002 |
| | $H_2$ | 0.55 | 0.491 | 0.338 | 0.427 | 1.241 | 0.664 | 0.698 | 0.651 | / | 0.158 |
| | $CO_2$ | 0 | 0 | 0.087 | 0.079 | 0.249 | 0.156 | 0.261 | 0.02 | / | 0.051 |
| Ratios | $C_1/C_2$ | 0.417 | 0.322 | 0.551 | 0.658 | 0.886 | 0.809 | 0.650 | 0.323 | / | 0.410 |
| | $C_1/(C_2 + C_3)$ | 0.259 | 0.208 | 0.351 | 0.377 | 0.626 | 0.577 | 0.422 | 0.209 | / | 0.267 |
| | $C_1/(C_1–C_5)$ | 0.179 | 0.151 | 0.239 | 0.260 | 0.366 | 0.344 | 0.269 | 0.151 | / | 0.189 |
| | $i$-C4/$n$–$C_4$ | 0 | 0 | 0.007 | 0.028 | 0.036 | 0.006 | 0 | 0 | / | 0.028 |
| | $C_{3ene}/C_3$ | 0.149 | 0.180 | 0.056 | 0.040 | 0.057 | 0.084 | 0.108 | 0.181 | / | 0.084 |
| | $C_{2ene}/C_2$ | 0.011 | 0.011 | 0.004 | 0.004 | 0.005 | 0.007 | 0.012 | 0.011 | / | 0.004 |

As for $i$-C5 and $n$-C5, their yields are too low to effectively determine $i$-C5/$n$-C5 ratios.

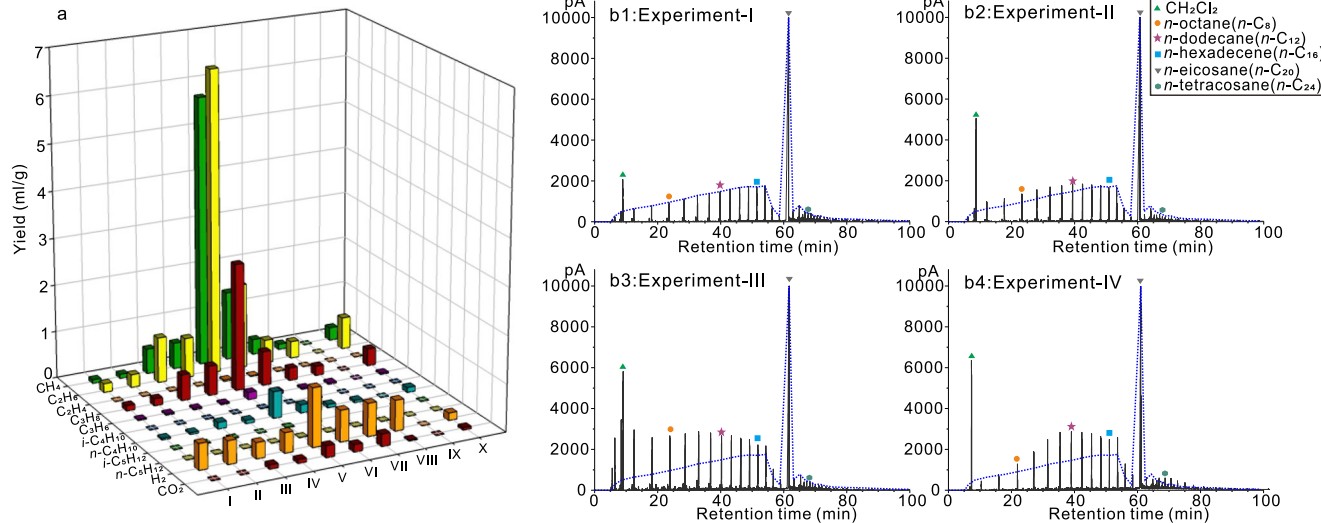

**Fig. 3 | Yields of gases and gas chromatograms of liquid hydrocarbons after thermal experiments. a** yields of $C_1$–$C_5$, $H_2$, and $CO_2$ in different thermal experiments. The analytical uncertainties for the yields of gas products had a relatively small error of <0.5% (see "Methods"). **b** gas chromatograms of liquid hydrocarbons in experiments I–IV, the dashed curves in (**b1**–**b4**) represent the pattern of the main compositions of the liquid hydrocarbons generated in experiment I with only $n$-$C_{20}H_{42}$. Detailed data for (**a**) listed in Table 1.

without $^{18}O$-labeled water, the $\delta^{18}O$ values of $CO_2$ are approximately −16.77‰, with an AT $^{18}O/^{16}O$ value of 0.1967 (Fig. 5b and Supplementary Table 4). In contrast, in experiments VI with $^{18}O$-labeled water ($C_{20}H_{42}$ + $D_2^{18}O$), the $\delta^{18}O$ value of the $CO_2$ reaches up to 95,731‰, with an AT $^{18}O/^{16}O$ value of 16.25, indicating that water functions as the oxygen source for $CO_2$ generation (Fig. 5b and Supplementary Table 4). In the hydrous system with feldspar (KAlSi$_3$O$_8$ with $\delta^{18}O$ of 9.4‰), its dissolution released oxygen (dominated by $^{16}O$) into water, forming hydroxyl ions (OH⁻)[46,47]. This process enriched the $^{18}O$ abundance of the original $^{16}O$-rich water, with a $\delta^{18}O$ increase from initial −4.97‰ to −1.17‰ and 0.23‰ of post-reaction waters in VII and VIII, respectively (Fig. 5b and Supplementary Table 5). In the $n$-$C_{20}H_{42}$-$D_2^{18}O$-feldspar system (V), the feldspar dissolution enriched the $^{16}O$

abundance of the $^{18}O$-labeled water. The oxygen released from feldspar dissolution was then assimilated into the newly formed $CO_2$, possibly through stepwise oxidation of alkanes by water[4]. This was supported by the lower $\delta^{18}O$ (AT $^{18}O/^{16}O$) value of $CO_2$ in the $n$-$C_{20}H_{42}$-$D_2^{18}O$-feldspar system (60,631‰/11.00 in Experiment V) compared to the $n$-$C_{20}H_{42}$-$D_2^{18}O$ system (95,731‰/16.25 in Experiment-VI) (Fig. 5b and Supplementary Table 4). Hence, in addition to water, feldspar may act as another source of oxygen for the $CO_2$ formed in the $n$-$C_{20}H_{42}$-$H_2O$-feldspar systems.

In experiment-VII with the inclusion of D-labeled $C_{20}D_{42}$, the post-reaction water exhibited a notably higher $\delta D$ (AT% D/H) value of 36,997‰ (0.5883) (Fig. 5c and Supplementary Table 5). Consistent with the isotopic data, the HF-NMR spectra of the post-reaction

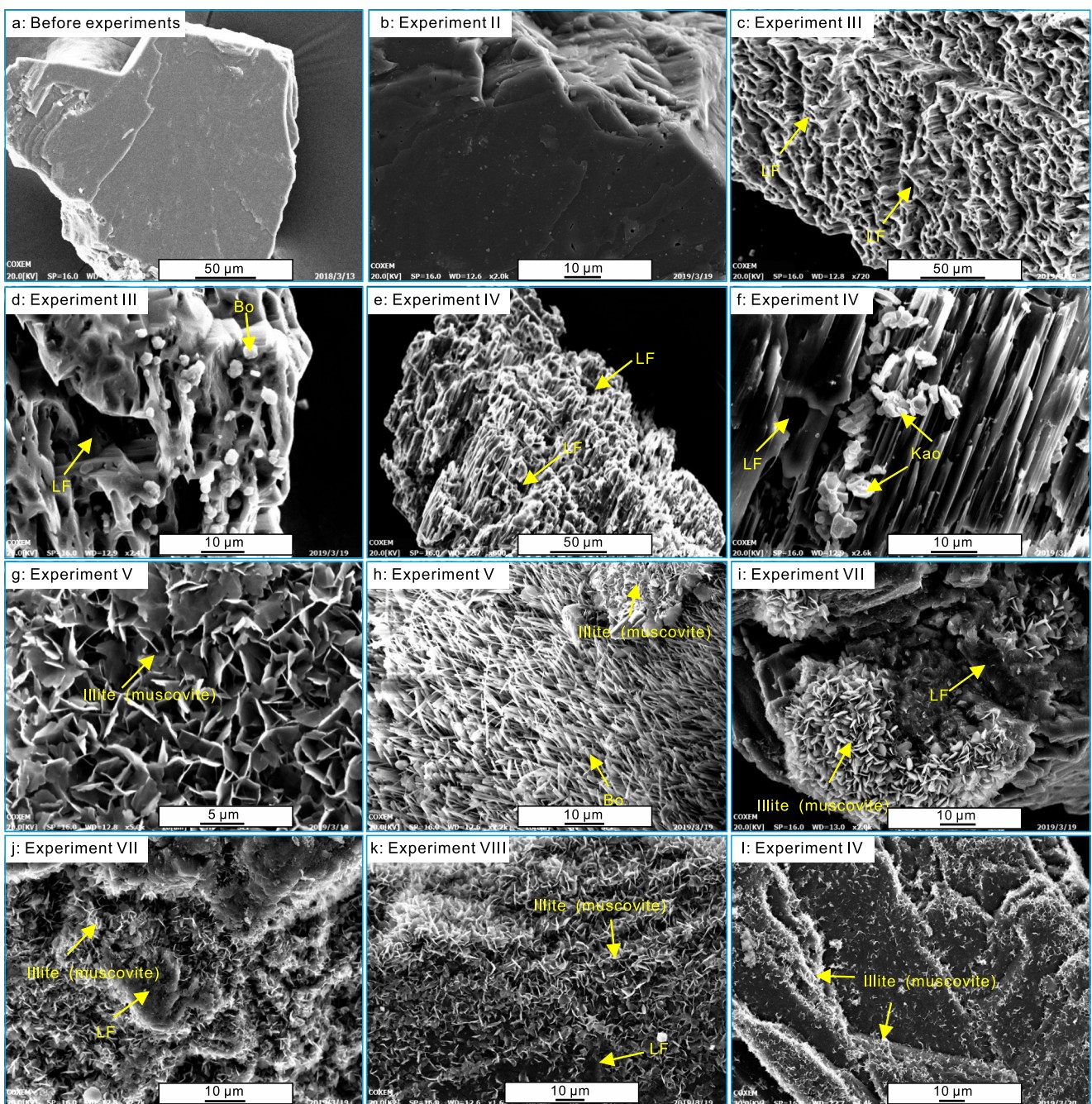

**Fig. 4 | Leaching of feldspars and precipitation of secondary minerals in thermal experiments.** SEM images of the original K-feldspar grains used in the experiments (**a**), leached K-feldspar and authigenic minerals after the 14-d experiments (**b–l**). **a** surface of the original K-feldspar grains. **b** surface of the K-feldspar after experiment in the anhydrous $n$-$C_{20}H_{42}$ + feldspar system (II). **c** extensively leached feldspar (LF) in the $C_{20}H_{42}$ + $H_2O$ + feldspar system (III). **d** extensively leached feldspar (LF) and euhedral boehmite (Bo) in the $C_{20}H_{42}$ + $H_2O$ + feldspar system (III). **e** extensively leached feldspar (LF) in the $C_{20}H_{42}$ + $D_2O$ + feldspar system (IV). **f** leached feldspar (LF) and euhedral kaolinite (Kao) in the $C_{20}H_{42}$ + $D_2O$ + feldspar system (IV). **g** flower-like illite (muscovite) aggregates in the $C_{20}H_{42}$ + $D_2{}^{18}O$ + feldspar system (V). **h** lash-shaped

boehmite in the $C_{20}H_{42}$ + $D_2{}^{18}O$ + feldspar system (V). **i, j** thin plate-shaped illite (muscovite) aggregates on the leached feldspar surface in the $C_{20}D_{42}$ + $H_2O$ + feldspar system (VII). **k** subhedral plate-shaped illite (muscovite) on the leached feldspar surface in the $C_{20}H_{42}$ + $H_2O$ + feldspar system (VIII). **l** small euhedral illite (muscovite) on the leached feldspar surface in the $H_2O$ + feldspar system (IX). In the systems with only 20 mg feldspar, the feldspars were leached quite extensively (**c–f**), and secondary minerals, including kaolinite and illite, were precipitated on K-feldspar surfaces and were also detected in the water solutions. In the systems with 2 g feldspars, the feldspar grains were dissolved, and illite and muscovite precipitated on the K-feldspar surfaces (**g–k**) and were also detected in the water solutions.

water also showed detectable deuterium content (Fig. 6b). Conversely, in experiments (VIII-X) without $C_{20}D_{42}$, the δD isotope composition of water after heating was significantly lower, with δD (AT% D/H) values of around −37‰ (0.014969) (Fig. 5c), and no deuterium signal was identified in the NMR spectra (Fig. 6b). The

significantly higher δD value (Fig. 5c) and notable D signal detected in HF-NMR spectra (Fig. 6b) of the post-reaction water in the $n$-$C_{20}D_{42}$-$H_2O$-feldspar system (VII) demonstrate the incorporation of $n$-$C_{20}D_{42}$-derived hydrogen (D) into the newly formed water molecules, forming D−O bounds. This phenomenon aligns with

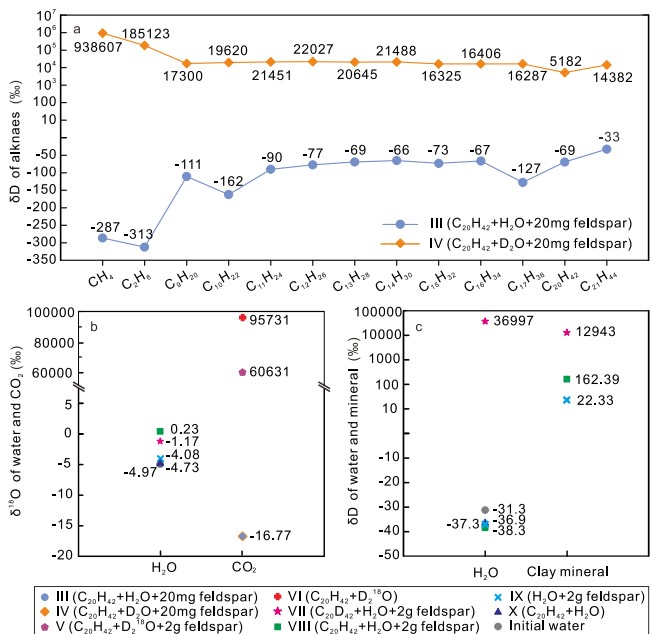

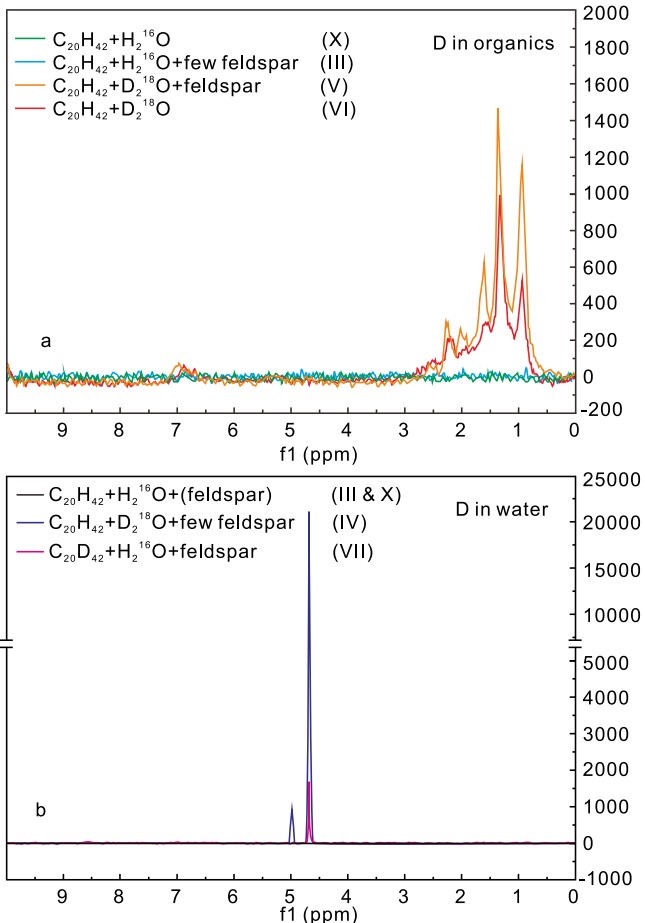

**Fig. 5 | Isotopic compositions of hydrocarbons, CO2, water, and clay minerals in the different anhydrous and hydrous systems with and without tracing isotope of D and 18O. a** δD of different gaseous and liquid hydrocarbons in hydrous systems with and without D₂O. **b** $^{18}$O of water and CO₂ in different hydrous experiments. **c** δD of water and clays in in different hydrous experiments. The analytical uncertainties for the determination of δD and $\delta^{18}$O were better than 2.0‰ and 0.1‰, respectively (see "Methods"). Detailed data are listed in Supplementary Tables 3–5.

**Fig. 6 | Deuterium (D)-NMR (nuclear magnetic resonance) of liquid oils and water after experiments. a** D-NMR of liquid organics in systems with and without D-labeled water. Peaks at 0.5–1.7 ppm, 1.7–2.5 ppm, and 6.7–7.0 ppm represent deuterium in alkanes, oxygen-containing organics (ketones), and organics with benzene rings, respectively; **b** D-NMR of water in systems with and without D-labeled *n*-eicosane. Detailed data have been deposited in Zenodo.

observation from previous studies in FSCTs, which illustrated a similar incorporation process when heated at 375 °C for 96 h (see Figs. 11 and 12 in ref. 21). In the anhydrous C₂₀H₄₂-feldspar system, *n*-C₂₀H₄₂-derived hydrogen does not react with feldspar to produce hydrogen-bearing minerals (Fig. 4b). Conversely, the presence of water in the feldspar-H₂O system lead to feldspar alteration reactions and precipitation of clay minerals (Fig. 4l). In system with only water and feldspar, water served as the hydrogen source for the newly formed hydrous minerals. In the system with alkane, after the transfer from alkane to newly formed water, the alkane-derived hydrogen was subsequently transferred to the newly formed clay minerals. This transfer is confirmed by the considerably higher δD value of the newly formed minerals in the *n*-C₂₀D₄₂-H₂O-feldspar system (12,943‰/0.219 in VII) in comparison to the *n*-C₂₀H₄₂-H₂O-feldspar (162‰/0.00018 in VIII) and H₂O-feldspar systems (22.33‰/0.00016 in IX) (Fig. 5c and Supplementary Table 5). Thus, both water and *n*-C₂₀H₄₂ served as hydrogen sources for the formation of OH-containing minerals in the hydrous systems with alkane, water and feldspar.

In summary, our experiments reveal additional processes beyond the previously reported H/O migration between alkanes and water[17,19,21], as well as the transfer of water-derived oxygen to CO₂[22,23]. Specifically, our findings demonstrate the migration of alkane-derived hydrogen to newly formed minerals, and the migration of feldspar-derived oxygen to newly formed oxygen-containing species, facilitated by the water medium. These observations highlight the absence of distinct boundaries between different organic and inorganic species in hydrous thermal systems, including alkanes and minerals.

## Microdroplet-induced pathway for interactions and mass transfer

For anhydrous experiments, previous studies have highlighted the dominance of the free radical mechanism in the degradation of crude oil

and pure alkanes[5,17,48]. Although catalytic hydrogenation reactions are plausible pathways due to the use of Hastelloy reactors, investigations have ruled out the catalytic impact of nickel-containing stainless steel, as experiments conducted in gold-line vessels yielded no measurable difference from those in stainless steel[17,19]. In our two anhydrous systems, the wide distribution of *n*-alkanes in the newly formed liquid hydrocarbons (Fig. 3b1, b2), the low ratios of *i*-C₄/*n*-C₄, and the presence of H₂ (Table 1) and some C₂₀₊ compounds further support the predominance of the free radical mechanism.

In our hydrous systems, the newly formed clay minerals (Fig. 4) with Brønsted acidic sites may enhance carbonium-ion reactions. However, ambient water, as demonstrated in many studies, tends to suppress the catalytic effect of clay minerals[4,10]. Even with addition of HCl, which significantly increases the H⁺ activity in low pH water (by five orders of magnitude), the yield of *i*-butane did not increase in thermal experiments (see Table 21 in ref. 17), suggesting that increasing concentrations of H⁺ and OH⁻ via water ionization at elevated temperature may not promote carbonium-ion reactions. While carbonium-ion reactions cannot be entirely excluded in our hot hydrous systems, the very low yields of *i*-C₄ and *i*-C₅, along with the low ratios of *i*-C₄/*n*-C₄ (0.03) (Table 1 and Fig. 3a), and the wide distribution of *n*-alkanes in the newly formed liquid hydrocarbons (Fig. 3b3–b4), indicate that the free radical mechanism predominantly governs the *n*-C₂₀H₄₂ degradation reactions in our hydrous systems. These results

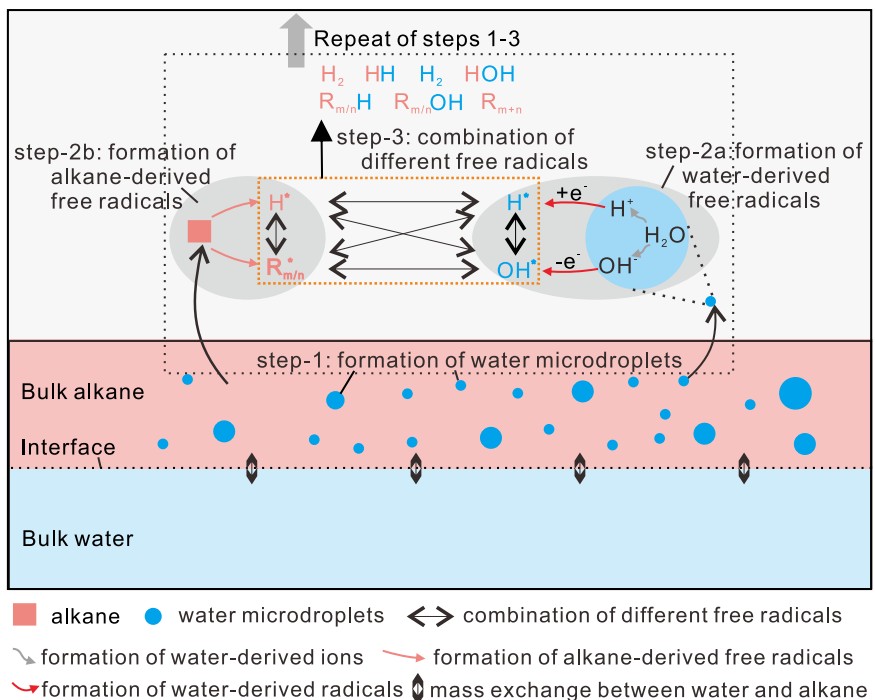

**Fig. 7 | Schematic diagram showing the pathways of microdroplet-induced interactions between alkane and water at elevated temperatures.** Step-1 represents the formation of water microdroplets in the alkane phase; step-2a represents the formation of water-derived free radicals based on the water microdroplets; step-2b represents the formation of alkane-derived free radicals; step-3 represents the recombination of different free radicals to form different species.

align with our previous hydrous experiments in both gold tubes and Hastelloy reactors[5,48] as well as other studies[17,19]. Furthermore, the yields of $C_1$–$C_3$ were four to five times higher in the hydrous systems compared to the anhydrous systems (Fig. 3a and Table 1), suggesting that the presence of water in the hydrous systems extensively promoted the free radical mechanism.

As hot bulk water cannot produce free radicals, we propose a microdroplet-induced pathway to elucidate the free radical reactions within the thermal hydrous systems. At high temperature, the ionization of water molecules in both the bulk water and water microdroplets generates numerous $H^+$ and $OH^-$ ions (Eq. (1)), with a significant increase in the water dissociation constant[4,36]. Strong electric fields (>$10^7$ V/cm), which induce the formation of water-derived free radicals, have been shown to occur at the interfaces of small water microdroplets with sizes less than 15 μm, as corroborated by both physical experiments[24] and theoretical studies[31,49]. In our experiments, many water microdroplets formed near the bulk alkane–water interfaces at elevated temperatures exhibit sizes smaller than 10 μm (Fig. 1; step-1 in Fig. 7). Although our current technology does not allow for direct measurement of the electric field on the moving individual water microdroplet in the hot FSCTs, the distinct OH* signal we obtained in the EPR spectra (Fig. 2a–c) demonstrates that the $OH^-$ and $H^+$ ions at the water microdroplet interfaces have been transformed to form hydroxyl free radicals (OH*) and hydrogen-free radicals (H*), respectively, via the release or acquisition of aqueous electrons (Eqs. (2–3))[24,29,31] (step-2a in Fig. 7). Simultaneously, the high temperature triggers formation of alkyl free radicals (R*) and H* from $n$-$C_{20}H_{42}$ and its intermediate products (Eqs. (4) and (9)); $m$, $n$, $x$, $y$ in the equations represent natural numbers greater than or equal to 1)[17,19,50,51], occurring in both the bulk alkane and the alkane microdroplets (step-2b in Fig. 7).

$$H_2O \rightleftharpoons H^+ + OH^- \tag{1}$$

$$OH^- \rightarrow OH^* + e^-_{(solv)} \tag{2}$$

$$H + e^-_{(solv)} \rightarrow H^* \tag{3}$$

$$C_{20}H_{42} \rightleftharpoons R^*_m + R^*_n + H^* \tag{4}$$

$$H^* + OH^* \rightarrow H_2O \tag{5}$$

$$R^*_m + H^* \rightleftharpoons R_mH_{2m+2} \tag{6}$$

$$R^*_m + R^*_n \rightleftharpoons R_{m+n} \tag{7}$$

$$H^* + H^* \rightleftharpoons H_2 \tag{8}$$

$$R_mH_{2m+2}(R_{m+n}) \rightleftharpoons R^*_x + R^*_y + H^* \tag{9}$$

$$R^*_m + OH^* \rightleftharpoons R_mOH \tag{10}$$

$$R_mOH \rightleftharpoons R_mO^* + H^* \tag{11}$$

$$R_mO^* + OH^* \rightleftharpoons R_mOOH \tag{12}$$

$$R_mOOH \rightarrow R_{m-1} + CO_2 \tag{13}$$

$$C_nH_{2n+2} \rightleftharpoons C_nH_{2n} + H_2 \tag{14}$$

$$H^+ + C_nH_{2n} \rightarrow C_nH^+_{2n+1} \tag{15}$$

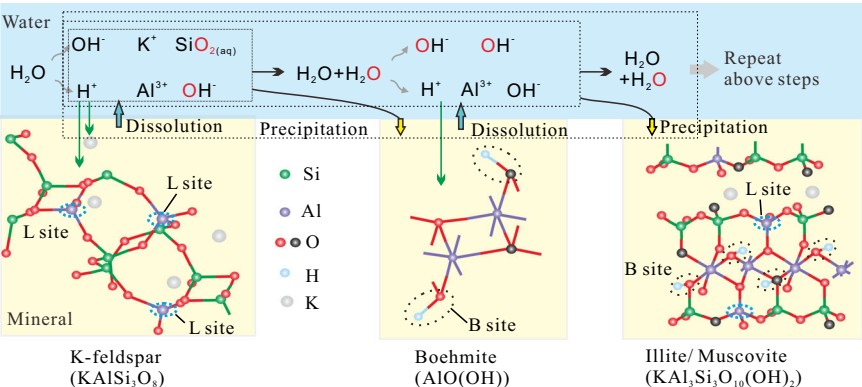

**Fig. 8 | Schematic diagram showing the pathways of mineral alteration in thermal systems with water and K-feldspar.** H$^+$ formed via water ionization reacts with feldspar, inducing various mineral alterations. The dissolution and precipitation of minerals like feldspar, boehmite, and illite / muscovite, along with the ionization and reformation of water molecules, entail the release or consumption of ions and solutes (H$^+$, OH$^-$, Al$^{3+}$, K$^+$, and SiO$_{2(aq)}$) into or from the water solution. These processes potentially lead to H/O exchange between water and minerals. The L and B sites represent the Lewis acid sites and the Brønsted acid sites in the aluminosilicate minerals, respectively (re-use permission has been obtained from Elsevier).

Once generated, the different free radicals interact within the microdroplets, at the interfaces, or within the bulk alkane solution, resulting in diverse recombination patterns (Eqs. (5–8)); step-3 in Fig. 7)[52]. OH$^*$ and H$^*$ recombination within and at the interfaces of the water microdroplets results in the formation of new water molecules (Eq. (5)). R$^*$ and H$^*$ recombination drives free radical thermal-cracking reactions, forming low-molecular-weight hydrocarbons (Eq. (6) and Fig. 3). While R$^*$ and R$^*$ recombination regulates free radical thermal-cross-linking reactions, leading to formation of high-molecular-weight hydrocarbons (Eq. (7) and Fig. 3b). The higher δD value of CH$_4$ compared to C$_2$H$_6$ in our study (Fig. 5a) supports the notion that most C−D bonds in the newly yielded alkanes were directly formed through continuous recombination of R$^*$ and H$^*$ derived from $n$-C$_{20}$H$_{42}$, its intermediate products and water. In addition, H$^*$ recombination results in the generation of H$_2$ (Eq. (8)). Unlike the hydrolytic disproportionation reactions catalyzed by iron-bearing minerals proposed by Seewald[4], the recombination of R$^*$ and water-derived OH$^*$ may also initiate stepwise oxidation reactions of alkanes, leading to the formation of oxygen-containing organic compounds (Eqs. (10–12))[30]. The decarboxylation of organic acids further occurs to form CO$_2$ (Eq. (13))[4,53]. The higher ratios of C$_1$/C$_2$, C$_1$/(C$_2$ + C$_3$), and C$_1$/(C$_1$−C$_5$), along with the lower ratios of C$_{2ene}$/C$_2$ and C$_{3ene}$/C$_3$ in the hydrous systems compared to the anhydrous systems (Table 1), suggest that the presence of additional water-derived H$^*$ may promote thermal-cracking reactions to form more low-molecular alkanes[5]. This promotion is likely due to an elevated opportunity for recombination between H$^*$ and low-molecular-weight R$^*$. Consequently, the ongoing formation and recombination of water-derived H$^*$ and OH$^*$, and $n$-C$_{20}$H$_{42}$-derived R$^*$ and H$^*$ (Eqs. (2–12)) at the interfaces of microdroplets within the water−alkane mixing zone likely initiate interactions between alkane and water, facilitating extensive transfers of hydrogen and oxygen among different organic compounds, water, CO$_2$, and H$_2$ (Fig. 7). In the alkane−water system, the disproportionation reactions of alkane also lead to the formation of some olefins (Eq. (14)), which may be converted to carbonium ions through simple proton addition (Eq. (15))[54]. However, with the additional supply of water-derived H$^*$, the production of olefins was significantly reduced when compared to the anhydrous systems, as evident by the low C$_{2ene}$/C$_2$ and C$_{3ene}$/C$_3$ ratios (Table 1). This decrease in olefin yield reduces the opportunities for the formation of 'carbonium-ion carbon atom' via the reaction between olefin and H$^+$ (Eq. (15)). This further explains the dominance of free radical reactions over carbonium-ion reactions in the thermal hydrous systems where water microdroplets are formed. Furthermore, the observed

formation of water microdroplets and the subsequent physico-chemical reactions expedite the degradation rate of $n$-C$_{20}$H$_{42}$ and increase yields of gases and liquid hydrocarbons in the hydrous systems (Fig. 3a, b3–b4 and Table 1). This aligns with the documented gradual yet evident interactions between oil and water in previously reported thermal hydrous systems without catalytic agents[32–34].

In the system with H$_2$O and feldspar, feldspar dissolution occurred, and illite (muscovite) precipitated on the feldspar surfaces (Fig. 4l). The reactions between water and feldspar have been extensively studies[47,55]. Following the formation of H$^+$ from ionization of water molecules (Eq. (1)), the H$^+$ reacts with feldspar, causing feldspar dissolution and the precipitation of secondary minerals (Eqs. (16–18))[55]. All these chemical reactions at each step involve the release or consumption of ions and solutes (H$^+$, OH$^-$, Al$^{3+}$, K$^+$, and SiO$_{2(aq)}$) into or from the water solution. Therefore, the continuous generation and recombination of H$^+$ and OH$^-$, as indicated by Eqs. (1, 16–18), potentially result in the H/O transfer between water and minerals (Fig. 8)[46,47,55].

$$KAlSi_3O_8(K\text{-feldspar}) + 4\,H^+ \rightleftharpoons 2H_2O + K^+ + Al^{3+} + 3SiO_{2(aq)} \quad (16)$$

$$Al^{3+} + 3OH^- \rightleftharpoons AlO(OH)(Boehmite) + H_2O \quad (17)$$

$$3Al^{3+} + 3SiO_2(aq) + K^+ + 10\,OH^- \\ \rightleftharpoons KAl_3Si_3O_{10}(OH)_2(illite/muscovite) + 4H_2O \quad (18)$$

$$R_mOOH \rightleftharpoons R_mOO^- + H^+ \quad (19)$$

$$CO_2 + H_2O \rightleftharpoons H^+ + HCO_3^- \quad (20)$$

In an alkane−water−feldspar system, the interactions between alkane and water result in the production of organic acids and CO$_2$ (Eqs. (10–13)), which dissolve in water, generating additional H$^+$ (Eqs. (19–20)) to promote the mineral alteration reactions (Eqs. (16–18))[4,5]. Simultaneously, the Lewis (L) acid sites presented in the feldspar and secondary minerals (Fig. 8) potentially facilitate the decarboxylation of water-soluble organic acids, leading to the formation of more low-molecular alkanes and CO$_2$[5,56]. Moreover, the micropores between feldspar grains could lead to the formation of

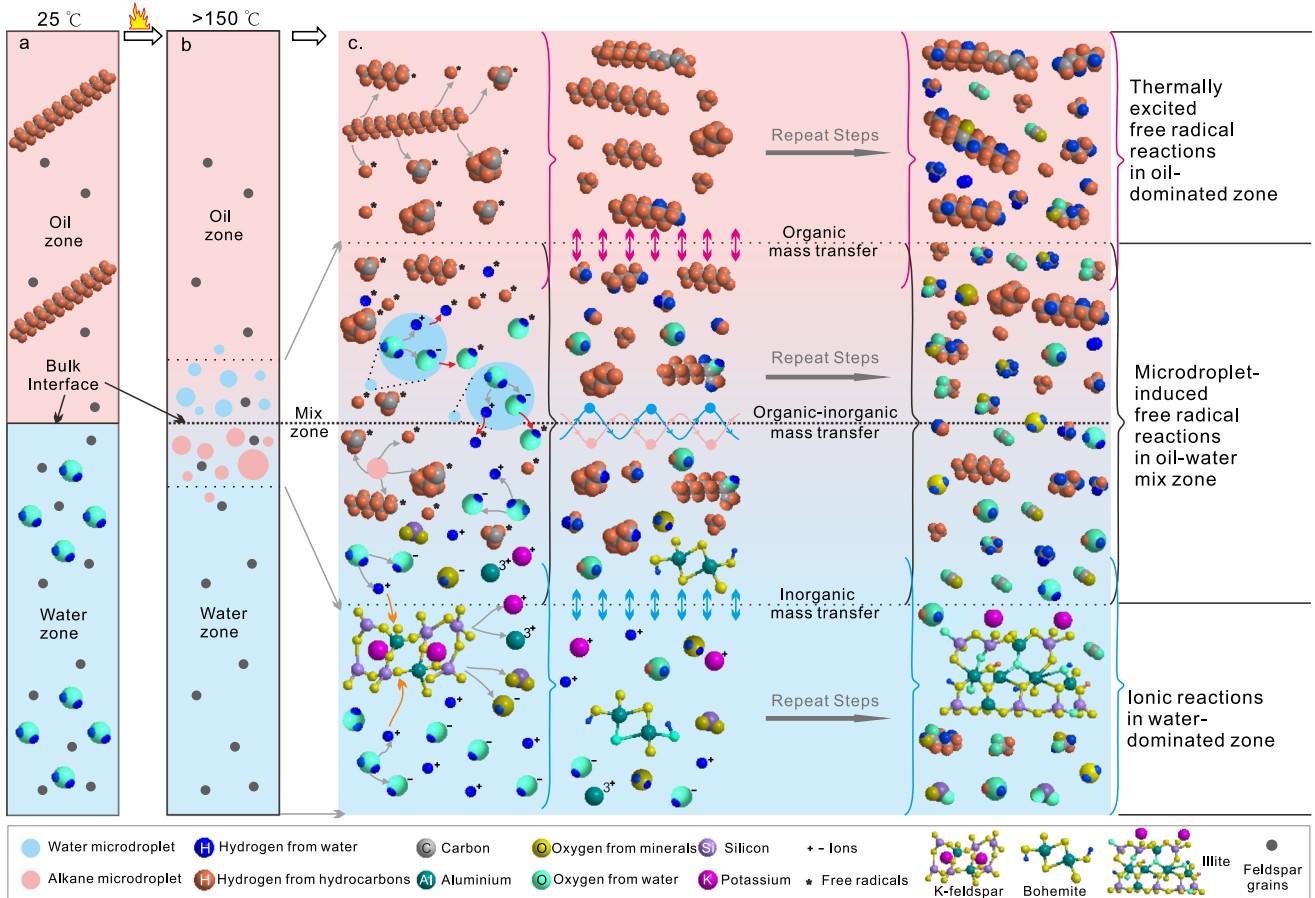

**Fig. 9 | Schematic diagram showing the microdroplet-induced model for organic-organic interactions and mass transfer among different species in thermal alkane–water-feldspar hydrous systems. a** Low temperature stystem without formation of microdroplets near the interface btween alkane and water zones. **b** Formation of microdeoplets near the alkane-water interface at elevated temperatures higher than 150 °C. **c** Occurrence of free radical reactions, ion reactions and mass exchanges in diffferent zones of the thermal alkane-water-feldspar system. In the water zone, water-derived ions react with minerals, lead to continuous mineral dissolution and precipitation, as well as H/O exchange between water and minerals. In the oil zone, high temperature facilitates the formation and recombination of alkane-derived free radicals, resulting in free radical thermal-cracking and cross-linking reactions. In the water–oil mix zone, water microdroplets form near the water–oil interface, trigging the formation of water-derived free radicals and initiating the organic–inorganic interactions between water and oil. The formation and recombination of water-derived and alkane-derived free radicals result in H/O exchange among water, alkanes, organic acids, $CO_2$, and $H_2$. These processes, with water serving as a bridge, erase the conventional boundaries between oil and minerals, facilitating the transfer of alkane/water/mineral-derived H/O among the newly formed organic and inorganic compounds.

additional alkane–water interfaces, contributing to the formation of small water microdroplets and further promoting interactions between the alkane and water. Thus, the continuous interplay of free radical reactions between alkane and water, combined with the concurrent ion reactions between water and minerals, provides a conceptual water-bridge model to elucidate the organic-inorganic interactions and associated mass transfers within thermal hydrous systems with alkane, water, and minerals (Fig. 9). This model involves the genesis of water microdroplets in the oil–water mixing zone, fostering active interfaces within these tiny reactors. Through mass circulation between microdroplets and bulk solutions, a connection is established between the physicochemical processes in both matrices. These simultaneous mechanisms erase the conventional boundaries that exist between organic and inorganic entities, facilitating the transfer of alkane/water/mineral-derived H/O among the newly formed alkanes, water, hydrous minerals, $CO_2$, and $H_2$. Our findings unveil the complex connection between the organic and inorganic compounds in thermal systems, intricately woven into the processes occurring in the myriad microenvironments near interfaces.

Our model, devoid of additional catalytic agents, offers an alternative elucidation for the observed, though gradual, interactions and mass transfer between organic and inorganic species in thermal hydrous systems[17–19,21]. It also accounts for the accelerated decomposition rates of organic species[5,32] and the prevalence of straight-chain alkanes in natural mature petroleum and light hydrocarbons[4,41]. These results offer a microscopic perspective on our understanding of deep sedimentary diagenetic and hydrous-metamorphic processes, as well as associated mass transfer cycles. For example, the observed temperature (150–165 °C) for the beginning formation of water microdroplets coincides remarkably with the critical threshold for extensive hydrocarbon degradation reported in natural hydrocarbon reservoirs[57,58]. This alignment suggests a potential link between the formation of water microdroplets and the onset of processes driving significant hydrocarbon degradation in subsurface environments. In addition to the debate of singularly organic and inorganic hypotheses for the genesis of deep methane in sedimentary basins and subduction zones[6,7,59–61], our study suggests that the formation of deep methane may involve both organic and inorganic species[16,18,20,51], with carbon derived from organic precursors (kerogen, oil, or pyrobitumen) and carbonate minerals, and hydrogen from organic species, water, and hydrous minerals[61–65]. Evidence for this can be seen in natural gas accumulations found in deeply buried reservoirs[66,67], and in low-molecular-weight hydrocarbons discovered in carbonated eclogite subduction zones[1]. Therefore, an in-depth investigation into the

multifaceted origins of deep hydrocarbons, encompassing both biotic and abiotic perspectives, is warranted, particularly when considering the contributions of inorganic-sourced hydrogen and carbon within the context of intricate localized source-to-sink relationships. In addition, apart from carbonate minerals, water and aluminosilicates may also serve as sources of oxygen for the generation of deep $CO_2$[13,23,68]. Furthermore, in conjunction with the frequently documented process of transferring carbon from organic sources to carbonate minerals[4], the hydrogen originating from organic compounds, in combination with water-derived hydrogen, could become integrated into hydrous minerals at elevated temperatures, implying a potential impact of surface-level biogeochemical processes on the deeper geosphere[69,70]. We are not intent to dismiss the catalytic models that may significantly accelerate organic–inorganic interactions. However, in geochemical systems where catalyst minerals are absent, our model may serve as a fundamental mechanism driving widespread organic–inorganic interactions in natural thermal geochemical systems.

## Methods

### Sample preparation

$n$-$C_{20}H_{42}$, $n$-$C_{20}D_{42}$, $H_2^{16}O$, $D_2^{16}O$, $D_2^{18}O$, crude oil and K-feldspar grains were used in this study. $n$-$C_{20}H_{42}$ was supplied by Aladdin Industrial Corporation (AIC, Shanghai) and had a purity greater than 99.5%; $n$-$C_{20}D_{42}$ was supplied by Canada C/D/N isotopes Inc., with 98.3% wt% deuterium; $H_2O$ was ultrapure water from AIC, with δD of −31.3‰ and $\delta^{18}O$ of −4.97‰. Pure $D_2^{16}O$ supplied by AIC was of 99.96 wt% deuterium. Pure $D_2^{18}O$ was supplied by the Wuhan Niuruide Special Gas Co., Ltd, with 99% deuterium and 97% $^{18}O$. The crude oil is a condensate oil from a deep hot reservoir in the Bohai Bay Basin, East China. Clean K-feldspar grains used in the experiments have sizes of 150 μm to 250 μm, and $\delta^{18}O$ of the feldspar is 9.4‰-SMOW. The K-feldspar grains were placed into Hastelloy alloy mesh bags (6 cm long) with a screen size of 150 mesh. Mesh bags were used to ensure the contact between minerals and water in the lower part and alkane in the upper part of the reactors (Supplementary Fig. 1).

### Isotope-tagged thermal experiments in Hastelloy reactors

The isotope-tagged thermal experiments were conducted in HTHP Hastelloy pressure reactors (20 mm outside diameter, 5-mm wall thickness, and 120 mm height) at the State Key Laboratory of Organic Geochemistry at the Guangzhou Institute of Geochemistry. All reactors were firstly heated at 750 °C for 8 h to burn any organic matter. Then, they were cleaned with acetone and distilled water and dried at 60 °C. Subsequently, $n$-eicosane ($n$-$C_{20}H_{42}$, $n$-$C_{20}D_{42}$), water ($H_2^{16}O$, $D_2^{16}O$, $D_2^{18}O$), feldspar grains, or mesh bags with feldspar grains were placed into the Hastelloy pressure reactors with different combinations of the species (Supplementary Table 1 and Supplementary Fig. 1). Once loaded, the open ends of the reactors were purged with argon to remove air from the reactor; subsequently, the reactors were sealed in the presence of argon. Lastly, the reactors were weighed to obtain the weight before heating. Then, the reactors were placed in a single furnace and heated at 340 °C (error <±1 °C) for 14 days. After heating, the reactors were quenched to room temperature in cold water within 10 min. After drying, the reactors were weighed again to ensure no leakage during the experiments. Experiments VII, VIII, and IX were repeated for five times to collect sufficient authigenic minerals for testing of δD composition.

### Analysis of gases, liquids, and minerals in Hastelloy reactors

We used a combination of compositional, isotopical, high-field nuclear magnetic resonance (HF-NMR), and mineralogical analyses to obtain the information on gas yields, isotopic compositions of gases, water and minerals, NMR spectra of liquid alkanes and waters, and textures of minerals.

After heating, the volatile components in the reactors were collected in a customized sampling device connected to an Agilent 6890 N gas chromatograph (GC) modified by Wasson-ECE Instrumentation[5]. The device was firstly vacuumed to $<1 \times 10^{-2}$ Pa. The reactor was then opened in the vacuumed device, allowing the gases to enter into the device. The valve connecting the device and the modified gas chromatograph was open to allow the gas components to enter the gas chromatograph; in this manner, the gas chromatographic analyses of the organic and inorganic gas components were performed using an automatically controlled procedure. The oven temperature for the hydrocarbon gas analysis was initially held at 70 °C for 6 min, ramped from 70 to 130 °C at 15 °C/min and from 130 to 180 °C at 25 °C/min, and was then held at 180 °C for 4 min, whereas it was held at 90 °C for the inorganic gas analysis. The analysis of all gases was carried out with a single injection. A calibration with external standard gases indicated that the amounts of the gas products measured using this device had a relatively small error of <0.5%. For testing of composition of the post-reaction liquid hydrocarbons, the liquid organics-water solutions were firstly separated using syringes to obtain the liquid hydrocarbons in the upper part of the reactors. After drying of the liquid hydrocarbons with anhydrous copper sulfate, the mixture of the liquid hydrocarbon with some $CH_2Cl_2$ was injected into an Agilent 6890 equipped with an HP-PONA column (50 m × 0.20 mm × 0.5 μm film thickness). The oven temperature was initially set at 35 °C with a hold time of 5 min and programmed to 70 °C at 3 °C/min, then to 300 °C at 4.5 °C/min with a final hold time of 35 min.

After the GC analysis, the remaining gases and liquid hydrocarbons were used for isotope analysis using a Thermo Fisher MAT-253 GC-isotope ratio mass spectrometry (GC-IRMS). For the liquid hydrocarbon sample obtained in system with $D_2O$, it was firstly washed three times using distilled water and then dried with anhydrous copper sulfate, for the purpose of reducing the impact of $D_2O$. This process was repeated until no color change of the ACS was visible. The analysis was performed on a VG Isochrom II interfaced to an HP 5890 GC fitted with a Poraplot Q column (30 mm × 0.32 mm i.d.). Helium (purity 99.999%) was used as the carrier gas. The column head pressure was set to 8.5 psi. For gas analysis, the gas chromatograph oven temperature was initially held at 50 °C for 3 min, ramped from 50 to 180 °C at 10 °C/min, and held at 190 °C for 5 min. The injection port temperature was held at 100 °C. For liquid hydrocarbon analysis, the gas chromatograph oven temperature was initially held at 60 °C for 1 min, ramped from 60 to 300 °C at 5 °C/min, and held at 300 °C for 5 min. The injection port temperature was held at 280 °C. The split ratio and injection volume were adjusted in time according to the content of each component. The pyrolysis furnace temperature was held at 1420 °C for the δD testing with ions ($m/z$ 2 and 3) monitored by IRMS. Standards of $CH_4$, $C_{16}H_{34}$ and $C_{27}H_{56}$ from Indiana University ($\delta^{13}C$ of −37.60‰ and δD of −41.3‰ for $CH_4$; $\delta^{13}C$ of −26.15‰ and δD of −9.1‰ for $C_{16}H_{34}$; $\delta^{13}C$ of −30.49‰ and δD of −17.28 for $C_{27}H_{56}$), and RM8564 $CO_2$ ($\delta^{13}C$ of −38.0‰ and $\delta^{18}O$ of −44.0‰) was used. The analytical uncertainties for the determination of δD and $\delta^{18}O$ were better than 2.0‰, and 0.1‰, respectively. The D/H and $^{18}O/^{16}O$ ratios were reported using direct atomic abundance ratios (AT D/H, AT $^{18}O/^{16}O$) and their delta values (δD, $\delta^{18}O$) in per mille (‰) relative to the $V_{SMOW}$. All samples were analyzed two to three times to ensure repeatability.

After separation of post-reaction water solutions in the lower part of the reactors. Each water sample was then filtered with bromoethane and activated carbon after low-temperature evaporation (80 °C) to remove the organic solutes (alcohols, organic acids, etc.). Finally, the water samples were purified using $C_{18}$ molecular sieves. The treated water samples were then analyzed using a LabRAM HR800 Raman spectrometer with a 532 nm laser excitation. No dissolved organic molecules were obviously detected in the water samples. After purification, the water samples were analyzed using IRMS to obtain the hydrogen and oxygen isotope composition. The D/H ratios were

reported using the AT D/H and the δD in unit of ‰ relative to the V$_{SMOW}$, and the $^{18}$O/$^{16}$O was reported using the delta values (δ$^{18}$O) in unit of ‰ relative to V$_{SMOW}$. The water standard GBW04403 (with δD −184.0‰ and δ$^{18}$O of −24.7‰) was used, and the analytical uncertainty of δD and δ$^{18}$O was less than 2.0‰ and 0.1‰, respectively.

For high-field nuclear magnetic resonance (HF-NMR) testing, the separated and processed liquid hydrocarbon and water samples were mixed with deuterium-free reagents to homogenize the field. The samples were then examined using an AVANCE III 600 MHz HF-NMR spectrometer with a broadband BBFO probe to obtain the $^1$H NMR and $^2$H NMR spectra of the samples. The standard ZG30 echo pulse sequence was used for the $^1$H NMR spectrum and the standard ZG echo pulse sequence was used for the $^2$H NMR spectrum. The pulse lengths were kept as short as possible to minimize any artifacts in the spectra due to finite pulse length effects. For the $^1$H NMR spectrum, the 90° pulse lengths were 10.5 μs and the echo delay was 40 μs. For the $^2$H NMR spectrum, the 90° pulses were 149 μs and the delay between pulses was 40 μs. In all experiments, the final delay prior to acquisition was set such that a few data points were collected before the top of the echo. This allowed us to manually correct the phase of the FID and shift the points in the time domain before removing the points before the top of the echo. This process is important for obtaining spectra with a flat baseline. At the beginning of each series, the sample set point temperature was raised to 25 °C, and the sample was allowed to equilibrate for at least 20 min. The $^1$H NMR spectrum was acquired four times, and the $^2$H NMR spectrum was acquired 32 times. At the end of each series, the temperature was set to the starting conditions of the series and the spectrum was collected again to verify the sample stability.

After the experiments, the feldspar grains were firstly cleaned using acetone and distilled water to remove the oil covering the mineral surfaces. The cleaned mineral grains were fixed on aluminum stubs with conducting tape and coated with gold. The minerals were identified using a Coxem-30plus scanning electron microscope (SEM) to describe the textures of the K-feldspar and to identify secondary minerals. A Bruker energy dispersive spectrometer (EDS) system (XFlasher Detector 430-M), which allows for the analysis at a specific spot of about 1 μm diameter was used to test the chemical composition of the minerals with an error of 0.1%.

The newly formed clay minerals which precipitated in the water solution were collected by centrifuging the water. The feldspar grains were cleaned using dichloromethane and distilled water to remove the oil covering the mineral surfaces. Clays precipitated on the feldspar grain surfaces were then separated gently by milling using a pestle and mortar. The fine debris removed from the feldspar grains was mixed with distilled water and stirred with a glass rod to collect the suspension in the upper layer. The suspension fluid was centrifuged at 492× $g$ to collect the precipitated clay minerals. The clays obtained from the water and feldspar surfaces were repeatedly cleaned using dichloromethane and distilled water five times to remove residual organic matter. The clay samples were identified using the SEM, EDS and Raman spectrometer to ensure no residuals of the organics. Lastly, the clays were dried and dispersed for isotope analysis. The prepared clays were analyzed using IRMS to obtain the hydrogen isotope composition. The standard IAEA-CH7 (δD = −100.3‰VSMOW) was analyzed before testing the samples, and the analytical uncertainty was less than 2.0‰. The D/H ratios were reported using the AT D/H and the δD and δ$^{18}$O in units of ‰ relative to the V$_{SMOW}$.

**In situ visual experiments in FSCTs**
The in situ visual experiments were conducted using a transparent silica capillary thermal experimental system equipped with microscope, camera, and video recording software (Supplementary Fig. 2), to ensure real-time observations of the formation and

evolution of microdroplets during the mixing of water and alkane/oil samples. Silica capillary tubes (TPS200794, with an inner diameter of 200 μm and an outer diameter of 794 μm) were used, with fluid assemblages of water-n-C$_{20}$H$_{42}$, water–liquid hydrocarbons after the n-C$_{20}$H$_{42}$ pyrolysis experiment, and water-crude oil (Supplementary Table 2 and Supplementary Fig. 3). First, water were injected into 25 cm long TPS200794 tubes with one end sealed, using a fine TPS075150 tube (inner diameter of 75 μm and outer diameter of 150 μm) connecting with a syringe. Then we inserted another fine tube with alkane/oil into the TPS200794 tubes at -10 mm from the bottom and slowly removed the fine tube to inject 10 mm alkane/oil. The open end of the TPS200794 tube was then connected to the experimental system, evacuated and then using constant-pressure liquid pump to fulfill the left part of the tube with water. After that, the sealed side of the tube was placed in a LinkamCAP-500 heating-cooling stage for heating. As the temperature increased from 25 °C to 350–410 °C and pressure from 0.1 MPa to -60 Mpa, the phenomena occurring adjacent to oil–water interfaces were recorded in detail.

**Electron paramagnetic resonance (EPR) spectrum measurements**
The EPR spectra were recorded at 413 K and 473 K using an EPR spectrometer (CIQTEK EPR200-Plus) with a continuous-wave X-band frequency (-9.6 GHz). The modulation field amplitude, frequency, and microwave power were set at 2 G, 100 kHz, and 2 mW (20 dB), respectively. Initially, the water sample was injected into a fused silica capillary tube (FSCT) with one end sealed for complete filling. Subsequently, a small quantity of the oil sample mixed with 5, 5-dimethyl-1-pyrroline-N-oxide (DMPO) was injected at various points into the water to create additional oil–water interfaces and promote microdroplet formation. The open end of the FSCT was then connected to a closed valve fitting and placed inside the EPR system protected by a quartz tube. After heating for 5 min, the EPR signals of each sample were collected three times to obtain average values. Then, background correction was performed on the collected data using the signal of an empty FSCT as a reference, followed by baseline correction and smoothing.

**Reporting summary**
Further information on research design is available in the Nature Portfolio Reporting Summary linked to this article.

## Data availability
The data of gas yields, gas ratios and isotopic compositions that support the findings of the study are included in the main text and supplementary information files. Data files of EPR are available in Figshare under accession code https://doi.org/10.6084/m9.figshare.24547492. The raw data of HF-NMR have been deposited in the Zenodo database at https://doi.org/10.5281/zenodo.11209873.

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

## Acknowledgements
The authors thank academician Zhijun Jin and professors Quanyou Liu, Wenxuan Hu, Xiaolin Wang for helpful discussions. This study is funded by the National Natural Science Foundation of China (No. 42222208 to G.Y., No. 41821002 to Y.C.), Science and Technology Innovation Project of Laoshan Laboratory (LSKJ202203402 to G.Y.), the Special fund for Taishan Scholar Project (No.tsqn201909061 to G.Y.), the Fundamental Research Funds for the Central Universities (20CX06067A to G.Y). Dr. Jin Guishan at Analytical Laboratory of BRIUG is thanked for the assistance in clay isotope tests; Dr. Liubin Feng at the High-field Nuclear Magnetic Resonance Research Center, Xiamen University is thanked for the assistance in HF-NMR tests; Dr. Fang Qing at CIQTEK Co., Ltd. is thanked for the assistance in EPR tests.

## Author contributions
G.Y. and Y.C. proposed the central idea and designed the experiments. G.Y. and Z.J. conducted the experiments. G.Y., Z.J., and X.H. performed the SEM, GC, isotope, HF-NMR, and EPR testing. G.Y., Y.C., H.-M.S., J.G., K.L., Z.J., and Y.W. interpreted the results and organized the writing of the manuscript. G.Y. and Z.J. wrote the manuscript. All authors contributed to the improvement of the manuscript.

## Competing interests
The authors declare no competing interests.
