## [Peer Review File · Nature Communications]

REVIEWER COMMENTS

Reviewer #1 (Remarks to the Author):

It remains mechanistically debated in governing the dynamics of hydrocarbons, water, minerals, CO₂, and H₂ within high-temperature rocks. In this submission, Yuan et al. observed the generation of water microdroplets at the water-oil interface at high temperatures, and ascribed the thermal cracking to water microdroplet-derived hydroxyl and hydrogen free radicals by designing comprehensive isotope-tagged experiments. Although they proposed possible mechanisms for the formation of low molecular weight hydrocarbons, it may be doubted for the contribution of water microdroplets considering the number of water microdroplets formed at the oil-water interface. I am also depressed by the missing of high molecular weight hydrocarbons (C₆ to C₂₀ and more). In addition, the formation of arenes and alkanes (C₁ to C₄) was absent in the mechanisms in equations 1-19.

Reviewer #2 (Remarks to the Author):

Review

This manuscript presents evidence that the recombination of water-derived and n-C₂₀H₄₂-derived free radicals drives the interactions between water and organic species, which triggers the interactions between water and minerals in thermal hydrous systems. Past considerations have invoked specific catalyst minerals to cause these chemical transformations, but as this manuscript points out, this view needs to be modified. I wholeheartedly agree with this point of view.

I believe however that the authors are missing two important references that relate to the arguments presented. The authors state in the introduction:

“With the occurrence of organic-inorganic interactions, the transfer of (a) water-derived H/O to kerogen, newly formed hydrocarbons 16-21 and oxygenated compounds 22,23, and (b) transfer of alkane-derived hydrogen to newly formed water molecules 21 have been observed in various thermal hydrous experiments. Even in studies without catalysts, the H/O transfer, involving the dissociation and regeneration of water molecules, has also been identified 18,21, suggesting a non-catalytic mechanism initiating interactions between organic species and hot water. However, non-catalytic interactions are still a matter of debate 5,24.”

I believe that two references need to be included and properly presented in the context of the above remarks:

Y. Xia, J. Li, Y. Zhang, Y. Yin, B. Chen, Y. Liang, G. Jiang, and R. N. Zare, “Contact Between Water Vapor and Silicate Surface Causes Abiotic Formation of Reactive Oxygen Species in an Anoxic Atmosphere,” *Proc. Natl. Acad. Sci. (USA)* 120, e2302014120 (2023).

and

X. Chen, Y. Xia, Z. Zhang, L. Hua, X. Jia, F. Wang, and R. N. Zare, “Hydrocarbon degradation by contact

with anoxic water microdroplets," J. Am. Chem. Soc. 145, 21538–21545 (2023).

I do not believe that the question of non-catalytic interactions remains a matter of debate. I suggest that it has been settled but perhaps not fully recognized by the geochemical community.

Once done, I will be pleased to review further this manuscript, which I do believe should be published.

The experimental results presented are truly wonderful.

Following are detailed response to each comment.

Comments from anonymous Reviewer #1:

1. It remains mechanistically debated in governing the dynamics of hydrocarbons, water, minerals, CO₂, and H₂ within high-temperature rocks. In this submission, Yuan et al. observed the generation of water microdroplets at the water-oil interface at high temperatures, and ascribed the thermal cracking to water microdroplet-derived hydroxyl and hydrogen free radicals by designing comprehensive isotope-tagged experiments. Although they proposed possible mechanisms for the formation of low molecular weight hydrocarbons, it may be doubted for the contribution of water microdroplets considering the number of water microdroplets formed at the oil-water interface.

Thank you very much for this very constructive comment, which encourage us to pursue additional experiments to provide more direct evidences. We conducted electron paramagnetic resonance (EPR) tests and utilized fluorescent indicators to study the behavior of water microdroplets formed at elevated temperatures near the alkane-water interfaces.

Fortunately, we obtained some promising EPR evidences (Line 123-132). We conducted the EPR tests to analyze free radicals using transparent fused silica capillary tubes (FSCTs) that contains 10-20 alkane/oil-water interfaces. Hydroxyl radicals (OH) were clearly detected in all the FSCTs containing water and three different alkane/oils at 200 °C when water microdroplets could be extensively formed (Fig. 2a-c). However, at 140 °C, which is below the critical temperature (150-165°C) for forming water microdroplets, no signal of hydroxyl radicals was obtained (Fig. 2a). These EPR spectra provide direct evidence to support our propose that water microdroplets behave differently from bulk water, leading to the formation of a large amount of free radicals that initiate extensive organic-inorganic interactions between water and alkanes (Page 11 Line 288-291).*

In nature deeply buried hydrocarbons reservoirs with a large number of micropores, numerous oil-water interfaces exist. In such cases, a large amount of microdroplets may be formed near these interfaces, which will have a significant impact on the evolution of crude oil.

2. I am also depressed by the missing of high molecular weight hydrocarbons (C₆ to C₂₀ and more).

Thank you for highlighting this point. In the revised version, we have included gas chromatograms (Fig. 3b1-b4) and isotopic composition data (Fig. 5a; Table S3) of the liquid hydrocarbons after thermal experiments. We use these data to discuss the impact of water on the generation of low/high-molecular-weight hydrocarbons, and also the transfer of water-derived hydrogen to liquid hydrocarbons (Line 138-140, 161-163, 190-194).

3. In addition, the formation of arenes and alkanes (C₁ to C₄) was absent in the mechanisms in equations 1-19.

We apologize for the oversight in not clearly describing this in the initial version. In the equations 5-15, the symbols m , n , x , y were used to represent natural numbers greater than or equal to 1, encompassing low molecular weight alkanes including C₁-C₄ and high molecular weight hydrocarbons including alkanes and arenes.

In the revised version, we have provided a description of m , n , x , y following the citation of Eq. 4 and 9 (Line 292-293). Thank you very much.

Comments from Reviewer :

1. This manuscript presents evidence that the recombination of water-derived and n-C₂₀H₄₂-derived free radicals drives the interactions between water and organic species, which triggers the interactions between water and minerals in thermal hydrous systems. Past considerations have invoked specific catalyst minerals to cause these chemical transformations, but as this manuscript points out, this view needs to be modified. I wholeheartedly agree with this point of view.

I believe however that the authors are missing two important references that relate to the arguments presented.

The authors state in the introduction: “With the occurrence of organic-inorganic interactions, the transfer of (a) water-derived H/O to kerogen, newly formed hydrocarbons¹⁶⁻²¹ and oxygenated compounds^{22,23}, and (b) transfer of alkane-derived hydrogen to newly formed water molecules 21 have been observed in various thermal hydrous experiments. Even in studies without catalysts, the H/O transfer, involving the

dissociation and regeneration of water molecules, has also been identified^{18,21}, suggesting a non-catalytic mechanism initiating interactions between organic species and hot water. However, non-catalytic interactions are still a matter of debate^{5,24}.”

I believe that two references need to be included and properly presented in the context of the above remarks:

Y. Xia, J. Li, Y. Zhang, Y. Yin, B. Chen, Y. Liang, G. Jiang, and R. N. Zare, “Contact Between Water Vapor and Silicate Surface Causes Abiotic Formation of Reactive Oxygen Species in an Anoxic Atmosphere,” *Proc. Natl. Acad. Sci. (USA)* 120, e2302014120 (2023).

X. Chen, Y. Xia, Z. Zhang, L. Hua, X. Jia, F. Wang, and R. N. Zare, “Hydrocarbon degradation by contact with anoxic water microdroplets,” *J. Am. Chem. Soc.* 145, 21538–21545 (2023).

Thank you very much for your positive comments and recognition of our work. The two inspiring references have been added as references 26 and 27 in the revised version.

In the revised version, we used the following to state the importance of water microdroplets with appropriate citations “Different from bulk water, water microdroplets at room temperature exhibit unique behavior^{24,25}. They produce hydroxyl radicals and hydrogen peroxide²⁵⁻³¹, facilitating the acceleration of organic reactions³²⁻³⁴, all without requiring additional catalyst. Chen et al (2023)²⁷ and Song et al (2023)³⁰ also reported the initiation of accelerated alkane degradation by contact of different alkanes with man-made water microdroplets at room temperature.” (Line 37-40)

Reference 26: Xia, Y. et al. Contact between water vapor and silicate surface causes abiotic formation of reactive oxygen species in an anoxic atmosphere. Proceedings of the National Academy of Sciences. 120, (2023).

Reference 27: Chen, X. et al. Hydrocarbon Degradation by Contact with Anoxic Water Microdroplets. J. Am. Chem. Soc. 145, 21538-21545 (2023).

(Line 591-594)

2. I do not believe that the question of non-catalytic interactions remains a matter of debate. I suggest that it has been settled but perhaps not fully recognized by the geochemical community. Once done, I will be pleased to review further this manuscript, which I do believe should be published. The experimental results presented are truly wonderful.

Thank you very much for your constructive comment and recognition of our work.

Yes, you are right. The non-catalytic interaction is not a problem when microdroplet are present. In the revised version, we updated the introduction and used the following to describe the existing problem:

“The transfer of water-derived H/O to kerogen, newly formed hydrocarbons¹⁶⁻²¹ and oxygenated compounds^{22,23}, as well as transfer of alkane-derived hydrogen to newly formed water molecules²¹, have been observed in thermal experiments. Notably, this H/O transfer, involving the dissociation and regeneration of water molecules, has also been identified in some thermal experiments without catalysts^{18,21}. This suggests the existence of a non-catalytic mechanism that may initiate interactions between organic species and hot water.

Different from bulk water, water microdroplets at room temperature exhibit unique behavior^{24,25}. They produce hydroxyl radicals and hydrogen peroxide²⁵⁻³¹, facilitating the acceleration of organic reactions³²⁻³⁴, all without requiring additional catalyst. Chen et al (2023)²⁷ and Song et al (2023)³⁰ also reported the initiation of accelerated alkane degradation by contact of different alkanes with man-made water microdroplets at room temperature. However, non-catalytic interactions in thermal hydrous systems are still a matter of debate in the geochemical community^{5,35}. Two schools of thought exist regarding this matter.....” (Line 32-42)

“In the realm of deeply buried, low-permeability rocks housing diverse geofluids, the formation of microdroplets becomes feasible, especially in high-temperature, high-pressure (HTHP) environments⁴⁴. These micro-environments may host intricate interactions among diverse organic compounds, water, minerals, and gases, with significant geological implications often overlooked^{27,30,32}. Motivated by recent advances in interfacial chemistry^{24-27,30,32}, we aim to investigate the formation, evolution and characteristics of water microdroplets near alkane-water interfaces at elevated temperatures, and to explore the basic physicochemical processes underpinning the organic-inorganic interactions in thermal geochemical systems from a microscopic perspective.” (Line 53-59)

REVIEWERS' COMMENTS

Reviewer #1 (Remarks to the Author):

The comments were addressed thoroughly, and hence I recommend its publication in the present form in NatCommun.

Reviewer #2 (Remarks to the Author):

I urge publication without further delay.